# The Drp1-Mediated Mitochondrial Fission Protein Interactome as an Emerging Core Player in Mitochondrial Dynamics and Cardiovascular Disease Therapy

**DOI:** 10.3390/ijms24065785

**Published:** 2023-03-17

**Authors:** Mulate Zerihun, Surya Sukumaran, Nir Qvit

**Affiliations:** The Azrieli Faculty of Medicine in the Galilee, Bar-Ilan University, Safed 1311502, Israel

**Keywords:** mitochondrial fission proteins, dynamin-related protein 1 (Drp1), mitochondrial fission 1 (Fis1), mitochondrial fission factor (Mff), mitochondrial dynamics 49 (Mid49), mitochondrial dynamics 51 (Mid51), protein–protein interactions (PPIs), mitochondrial dynamics, cardiovascular diseases (CVDs), mitophagy, protein structure, fusion, fission

## Abstract

Mitochondria, the membrane-bound cell organelles that supply most of the energy needed for cell function, are highly regulated, dynamic organelles bearing the ability to alter both form and functionality rapidly to maintain normal physiological events and challenge stress to the cell. This amazingly vibrant movement and distribution of mitochondria within cells is controlled by the highly coordinated interplay between mitochondrial dynamic processes and fission and fusion events, as well as mitochondrial quality-control processes, mainly mitochondrial autophagy (also known as mitophagy). Fusion connects and unites neighboring depolarized mitochondria to derive a healthy and distinct mitochondrion. In contrast, fission segregates damaged mitochondria from intact and healthy counterparts and is followed by selective clearance of the damaged mitochondria via mitochondrial specific autophagy, i.e., mitophagy. Hence, the mitochondrial processes encompass all coordinated events of fusion, fission, mitophagy, and biogenesis for sustaining mitochondrial homeostasis. Accumulated evidence strongly suggests that mitochondrial impairment has already emerged as a core player in the pathogenesis, progression, and development of various human diseases, including cardiovascular ailments, the leading causes of death globally, which take an estimated 17.9 million lives each year. The crucial factor governing the fission process is the recruitment of dynamin-related protein 1 (Drp1), a GTPase that regulates mitochondrial fission, from the cytosol to the outer mitochondrial membrane in a guanosine triphosphate (GTP)-dependent manner, where it is oligomerized and self-assembles into spiral structures. In this review, we first aim to describe the structural elements, functionality, and regulatory mechanisms of the key mitochondrial fission protein, Drp1, and other mitochondrial fission adaptor proteins, including mitochondrial fission 1 (Fis1), mitochondrial fission factor (Mff), mitochondrial dynamics 49 (Mid49), and mitochondrial dynamics 51 (Mid51). The core area of the review focuses on the recent advances in understanding the role of the Drp1-mediated mitochondrial fission adaptor protein interactome to unravel the missing links of mitochondrial fission events. Lastly, we discuss the promising mitochondria-targeted therapeutic approaches that involve fission, as well as current evidence on Drp1-mediated fission protein interactions and their critical roles in the pathogeneses of cardiovascular diseases (CVDs).

## 1. Introduction

Mitochondria are the cell’s “powerhouses”, generating adenosine triphosphate (ATP) via oxidative phosphorylation to meet cellular energy demands [1,2]. Being key metabolic hubs, mitochondria also play crucial roles in a plethora of other biological processes, including cell signaling, protein and nutrition transport across the cell, fatty acid oxidation, cell differentiation, apoptosis, and calcium signaling, among others [3]. Mitochondria are the only subcellular organelles in animal cells that harbor their own genomes/circular DNA of maternal origin [4]. Mitochondria sustain normal physiological functions by maintaining a steady state or balanced interplay between two contradictory fission and fusion events to maintain a healthy mitochondrial network. Fusion and fission events occurring concurrently determine the overall form, size, and population of mitochondria by regulating mitochondrial dynamics [5]. Mitochondrial fusion is the process of union or merging of two mitochondria to produce a distinctly healthy mitochondrion. On the other hand, the process by which a single mitochondrion may divide into two or more daughter organelles to eliminate the damaged fragmented mitochondria is referred to as mitochondrial fission [6,7]. Therefore, biological systems need highly sophisticated balanced control mechanisms correlating mitochondrial dynamics and mitochondrial quality-control mechanisms to maintain the fitness of mitochondrial pools and networks.

Mitochondrial fission is habitually regulated by cytosolic GTPase dynamin-related protein 1 (Drp1) [8,9]. The recruitment of Drp1 to mitochondria is facilitated by an array of specific adaptor proteins located in the outer mitochondrial membrane (OMM), such as mitochondrial fission protein 1 (Fis1), mitochondrial fission factor (Mff), and mitochondrial dynamics proteins of 49 and 51 kDa (Mid49 and Mid51 (also known as mitochondrial elongation factor 2 (MIEF2) and mitochondrial elongation factor 1 (MIEF1), respectively)). During mitochondrial fission, adaptor protein-mediated translocation of Drp1 to the OMM occurs and Drp1 binds at the pre-constricted sites of contact with the endoplasmic reticulum (ER). Drp1 further undergoes oligomerization to form a helical structure in the OMM in a GTP-dependent manner which encircles, constricts, and cleaves the mitochondrion into two daughter mitochondria [10,11,12]. The translocation of cytosolic Drp1 is mediated by its interaction with OMM adaptor proteins. Together, they form the core of the mitochondrial fission machinery, which is regulated by post-translational modifications (PTMs), including ubiquitination, SUMOylation, phosphorylation, S-nitrosylation, and acetylation [8]. However, the adaptor protein-mediated Drp1 interaction networking and molecular mechanisms behind the Drp1-mediated fission interactome remain vague and need further validation.

Mitochondria are the primary ATP producers of the heart, liver, kidney, and brain. The heart, as the most energy-demanding organ, is profoundly reliant on mitochondrial ATP production to power contractile function and cardiomyocyte metabolism [9,10]. The disrupted mitochondrial dynamics and quality-control mechanisms together play a critical role in the pathophysiology of numerous cardiovascular diseases (CVDs), impacting cellular energy, reactive oxygen species (ROS) production, intracellular calcium levels, and the production of apoptotic signals [11]. In this context, these pleiotropic roles point to clear-cut evidence of correlation between fission-mediated mitochondrial dynamics and CVDs. As a result, we will also discuss the current evidence on Drp1 and Drp1-mediated fission protein interactions and their critical roles in the pathogeneses of CVDs, such as pulmonary arterial hypertension (PAH), heart failure, cardiac hypertrophy, and myocardial infarction. This review provides current perspectives on the emerging evidence on structural elements of key fission proteins and supportive roles of adaptor proteins, Drp1-mediated adaptor protein–protein interactions (PPIs), regulation of mitochondrial fission proteins, and the involvement of fission proteins in CVDs. Overall, we focus on emerging trends in the molecular understanding of Drp1-mediated mitochondrial fission, the associated protein interactions (interactome), and their regulatory role in fission and mitochondrial dynamics as potential novel targets for treating CVDs in the future.

## 2. Mitochondrial Fission

Mitochondrial fission is a multi-step process which ensures the division of mitochondria to eliminate damaged mitochondria controlled mainly by the large GTPase, Drp1, and a cascade of other adaptor proteins [12]. Many of these process modifications are connected to mitochondria’s capacity to engage in the highly coordinated processes of fission (division of a single organelle into two or more independent structures) [13]. Mitochondrial fragmentation is often triggered during mitochondrial dysfunction mainly associated with elevated stress levels and apoptosis. Drp1 is an evolutionarily conserved protein that was first described in *Caenorhabditis elegans* [14] and yeast [15] before being extensively studied in mammals [16]. It is a cytosolic protein that is dynamically recruited to mitochondrial and peroxisomal membranes, where it is oligomerized and promotes membrane constriction in a GTP-dependent manner [17]. The genetic deletion of Drp1 in a variety of cell lines and animal models causes significant elongation of both mitochondria and peroxisomes [18,19]. Fission is mediated by Drp1 and interaction between the mitochondria and the endoplasmic reticulum (ER). However, the entire mechanism underlying Drp1 recruitment to mitochondria remains unknown, despite previous findings having implicated the involvement of actin polymerization via ER-localized inverted formin 2 (INF2) protein [20]. Mitochondrial fission preferentially occurs at ER contact sites, with the ER circumscribing mitochondria (Figure 1) [21].

Drp1-dependent mitochondrial fission is often triggered through mitochondrial dysfunction during elevated stress levels and apoptosis. It is a multi-functional process which involves four crucial steps: (i) Drp1 translocation to the OMM, (ii) subsequent higher-order assembly, (iii) GTP hydrolysis, and, finally, (iv) disassembly [22]. Drp1 interacts with receptors in the OMM, generating a functional complex and transport from the cytoplasm to the fission site [23,24,25]. Drp1 then oligomerizes at the OMM, forming a spiral that encircles the mitochondrion and mediates its scission. Drp1-mediated mitochondrial fission appears to be initiated by the early constriction of mitochondria after contact with the ER [21] via the association of the ER-associated INF2 (a formation that accelerates both actin polymerization and depolymerization) and the actin component of the cytoskeleton [26]. Several research studies are focusing on Drp1 PTMs in fission. According to recent findings, Drp1 controls the dynamics of the fission process by moving from the cytosol to the OMM, where it interacts with other proteins that are part of the fission machinery, such as Fis1, Mff, Mid49, and Mid51 [27,28]. Table 1 presents critical information about the locations and molecular functions of major mitochondrial fission proteins. Understanding the structure and composition of fission proteins is critical for future research findings and for understanding the roles of each protein.

### 2.1. Proteins Involved in Mitochondrial Fission

#### 2.1.1. Structural Elements and Roles of Drp1 in Mitochondrial Fission

Drp1 consists of four domains: a GTPase effector domain at the C-terminus, a variable domain (also known as insert B), a helical middle domain, and a highly conserved GTPase domain at the N-terminus (Figure 2) [57]. The protein domain structure of Drp1 revealed that Drp1 consists of a head (a GTPase domain) sitting on a neck region known as the bundle signaling element, a stalk containing the middle domain and GTPase effector domain (GED), as well as a foot of the uncharacterized variable domain (VD) (Figure 2) [58]. The PTMs of Drp1 in the VD and GED domains are depicted in Figure 3. Furthermore, the crystal structure revealed that the variable domain functions as a hinge by forming a T-shaped dimer or tetramer and effectively binding the targeted membrane (PDB: 3W6O [59]). Drp1 can only interact with its receptor to produce a functional complex that is then aggregated into an oligomer and transported to fission sites because it lacks a pleckstrin homology domain that interacts with lipids [60]. Four OMM receptors and/or adaptors have been identified to date that recruit Drp1 from the cytosol to the OMM for fission: Fis1, Mff, Mid49, and Mid51. Fis1 was the first OMM adaptor to be identified as a Drp1 recruiter [61]; it generates oligomers on the OMM that act as scaffolds and interacts with Drp1 via two-tetratricopeptide repeat-like motifs. While Drp1/Fis1 PPI is critical for mitochondrial fission and homeostasis, excessive Drp1/Fis1 interaction, on the other hand, has been linked to some pathological conditions, including Parkinson’s and Huntington’s diseases [61]. Mff is an essential factor in mitochondrial recruitment of Drp1 and was demonstrated to be critical for mitochondrial recruitment of Drp1 during mitochondrial fission [62]. Mid49 and Mid51 can influence Drp1/Mff interactions as well as Drp1 accumulate on mitochondria [63]. Overexpression of heterologous Mid49 and Mid51 on the OMM promotes fusion by sequestering inactive Drp1; conversely, increased endogenous Mid49 and Mid51 optimizes OMM scission [25].

##### Regulation of Drp1 via Post-Translational Modifications

The prime factor in the fission process is the recruitment of Drp1 from the cytosol to the OMM, and it is regulated by a series of PTMs, including phosphorylation, SUMOylation, ubiquitination, and S-nitrosylation, which alters the oligomerization property of Drp1 (Figure 3) [65]. Although each of these changes will be detailed separately, it is likely that at least some of them occur simultaneously and that changes at one site may have an impact on changes at other sites. The adaptors Fis1, Mff, Mid49, and Mid51 recruit Drp1 monomers to the OMM, allowing helical ring Drp1 oligomers to form. As a result of GTP hydrolysis by Drp1, the OMM constricts and scission occurs. This activation of mitochondrial fission is linked to several PTMs within the VD and the GED of Drp1.

Regulation of Fission by Drp1 Phosphorylation

Phosphorylation is the process of phosphorylating a chemical compound either by reaction with an inorganic phosphate or by the transfer of a phosphate from another organic phosphate. Phosphorylation is one of the most well-studied mitochondrial fission regulators. Characterized PTMs of Drp1 act both as activators and inhibitors depending on the site of modification. Ser40, Ser44, Ser579, Ser585, Ser592, Ser616, Ser637, Ser656, and Ser693 are only some of the phosphorylation sites that have been identified for Drp1 [66]. Two important sites, Ser616 and Ser637, have been extensively studied. Ser616 phosphorylation promotes mitochondrial fission and localization of Drp1 in the OMM [67], whereas phosphorylation of Ser637, which is phosphorylated by protein kinase A (PKA, also known as cyclic AMP-dependent protein kinase) and dephosphorylated by calcium-dependent phosphatase calcineurin (CaN), could be an inactivating step reducing mitochondrial fission [67,68].

Phosphorylation of Ser616 is likely to activate fission because it promotes binding to other fission proteins, whereas phosphorylation of Ser637 could be an inactivating step [69]. Likewise, Ser656 is phosphorylated by PKA and dephosphorylated by calcineurin and the neuron-specific Bβ2 regulatory subunit of protein phosphatase 2A (PP2A), resulting in mitochondrial fission regulation. Additionally, glycogen synthase kinase-3 beta (GSK3β) can regulate Ser693 phosphorylation [70]. Several protein kinases, including Rho-associated protein kinase (ROCK), protein kinase *C* (PKC), cyclin-dependent kinase 1 (CDK1), extracellular signal-regulated protein kinase 1/2 (ERK1/2), and calmodulin-dependent protein kinase II (CaMKII), are known to phosphorylate Ser616 in several types of cells [71]. CDK-catalyzed phosphorylation has been shown to have a reciprocal effect on Drp1 oligomerization and translocation [11]. Moreover, a recent study proposed a new pathway by which Ser616 phosphorylation is regulated in a RhoA/ROCK-dependent manner in cardiomyocytes to transmit environmental signals to mitochondria [72]. Even though Ser637 phosphorylation is mostly associated with reduced Drp1 GTPase activity, the exact function of Ser637 phosphorylation is influenced by a variety of external and internal factors, such as cell type, the Drp1 receptors, cellular context, and upstream molecules [73]. For instance, phosphorylation of Ser637 by PKA inhibits the interaction between the GTP-binding domain and the GED domain, resulting in reduced GTPase activity and reduced mitochondrial fission [30]. In hepatic ischemia/reperfusion (I/R), calcineurin-induced dephosphorylation of Drp1 at Ser637 contributes to Drp1 translocation to OMM [62]. Another study involving cardiac mitochondrial signaling systems found that protein kinase D (PKD)-mediated phosphorylation at Ser637 promotes mitochondrial fragmentation under pathophysiological circumstances [74].

It has been shown that Drp1 is translocated to mitochondria in hepatic I/R when calcineurin dephosphorylates it at Ser637 [62]. PKD-mediated phosphorylation at Ser637 was observed under pathophysiological conditions to promote mitochondrial fragmentation in another study involving cardiac mitochondrial signaling mechanisms [74].

Regulation of fission by Drp1 S-nitrosylation

S-nitrosylation is a process by which nitric oxide (NO) is covalently conjugated to Cys residues of target proteins. It has been found that S-nitrosylated Drp1 accumulates in neuronal cells and is associated with increased Drp1 oligomerization, altered protein conformation, and increased GTPase activity [66]. S-nitrosylation of Drp1 at Cys644 within the GED domain, caused by NO-facilitated redox-mediated mechanisms, can result in mitochondrial fragmentation, synaptic injury, and bioenergetics failure that may be associated with Alzheimer’s disease [75]. S-nitrosylated Drp1 was found to be crucial in the excessive mitochondrial fission brought on by mutant huntingtin in both animal models and humans with Huntington’s disease [76]. However, the findings are controversial, since other groups have shown that Drp1 S-nitrosylation does not affect Drp1 activity but does promote the phosphorylation of Ser616, which increases fission [75]. The results show that aberrant NO production results in mitochondrial and synaptic dysfunction, although S-nitrosylation of Drp1 may not be the only critical precursor. Another study found that the mechanism underlying NO-induced mitochondrial fragmentation in Alzheimer’s disease is not regulated by Drp1 S-nitrosylation because the levels of S-nitrosylated Drp1 in aged brains do not affect Drp1 activity [75]. Instead, NO-induced mitochondrial fragmentation in Alzheimer’s disease was discovered to be regulated by Drp1 phosphorylation at Ser616 [77]. As the number of publications concerning Drp1 S-nitrosylation is limited, further studies are required to confirm its functional relevance.

Regulation of Fission by Drp1 Ubiquitination

Ubiquitin is a 76-amino acid protein that modifies the biological functions of proteins by either altering their functional connections or PTMs them for further proteasomal degradation or modulates their biological functions (proteins control gene transcription, DNA repair and replication, intracellular trafficking, and virus budding) [78]. Drp1 has been discovered to be ubiquitinated by Parkin, an E3 protein ligase that targets Drp1 for proteasomal degradation and hence regulates the fission–fusion process in mitochondria [79]. Parkin knockdown or pathogenic mutations are linked to decreased Drp1 degradation, which leads to increased Drp1 activity and excessive mitochondrial division, which can lead to various diseases, such as Parkinson’s disease [79,80,81]. Another E3 ubiquitin ligase involved in mitochondrial dynamics is mitochondrial membrane-associated RING finger E3 ubiquitin ligase, also known as MARCH5. Drp1 is polyubiquitinated by MARCH5, which further guides Drp1 for proteasomal degradation [10].

Drp1 accumulation was also detected in MARCH5-deficient mice’s embryonic fibroblasts and organs, along with uncontrolled mitochondrial division and tremendous increase in ROS [48]. In addition to the MARCH5-Drp1-UPS (ubiquitin-proteasome system) pathway, MARCH5 can affect mitochondrial homeostasis in the MARCH5-mediated mitophagy pathway. When MARCH5 was re-expressed in MARCH5-knockout cells, the inhibition of stress-induced apoptosis was reversed, resulting in normal fission [62]. Drp1 is also involved in regulation of MARCH5 activity [78]. Cherok et al. also elucidated the combined regulatory roles of Mff and Drp1 in E3 ubiquitin ligase MARCH5-dependent degradation of MiD49. Knockouts of either fission proteins Drp1 or Mff led to reduced expression and increased ubiquitination of MiD49 [82]. Furthermore, mitochondrial accumulation of Drp1 and decreased cellular mobility of Drp1 in cells expressing MARCH5 ring mutants suggest that MARCH5 activity regulates Drp1 subcellular trafficking, most likely by influencing correct assembly at scission sites or the disassembly step of fission complexes [83]. Recent research will hopefully shed light on the role of MARCH5 in Drp1 activity and mitochondrial function.

Regulation of Fission by Drp1 SUMOylation

SUMOylation is a reversible and dynamic alteration that changes the subcellular localization of proteins or shields them from ubiquitin-mediated degradation. Small ubiquitin-like modifier (SUMO) proteins are also important in Drp1 activity regulation and persistent binding of Drp1 to the OMM, resulting in enhanced mitochondrial fission [84]. Mitochondrial-anchored protein ligase (MAPL) SUMOylates Drp1, marking numerous non-conserved lysine residues inside its middle domain (e.g., Lys532, Lys535, Lys558, Lys568, Lys594, Lys597, Lys606, and Lys608). During apoptosis, MAPL-induced SUMOylation was detected in ER/mitochondrial contact sites, helping to stabilize an ER/mitochondrial signaling platform involving cristae remodeling, calcium flow, and mitochondrial constriction [62]. Zinc-induced cardioprotection against I/R damage is mediated via SUMOylated Drp1. Mechanistically, mitophagy is induced in response to Drp1 SUMOylation, which inhibits ROS production, eliminates damaged mitochondria, and regulates mitochondrial quality, resulting in enhanced cardiac performance and reduced I/R damage [40]. Sentrin/SUMOylation-specific protease 5 (SENP5) is required for Drp1 deSUMOylation and mitochondrial fission suppression via its association with Drp1. SENP5 is the only SENP that is elevated in heart failure in humans. In cardiomyocyte-specific SENP5-overexpressing mice, reduced SUMOylation of Drp1 promotes larger mitochondria and cardiomyocyte apoptosis, culminating in cardiomyopathy and heart failure. Additionally, it has been reported that SENP2 and SENP3 removed Drp1 SUMOylation. However, Drp1 SUMOylation catalyzed by different SUMO isoforms leads to different functional effects. For example, Drp1 that has been SUMO1-modified is preferentially localized to mitochondria, whereas the recruitment to mitochondria of Drp1 that has been SUMO2/3-modified is repressed [43]. Further research into the involvement of various SUMO isoforms in Drp1-mediated mitochondrial fission is needed [85].

#### 2.1.2. Structural Elements and Roles of Fis1 in Mitochondrial Fission

Mammalian Fis1 is the ortholog of yeast Fis1 and was first identified as the receptor for Drp1 recruitment to mitochondria in mammals. Mammalian Fis1, like yeast Fis1, is localized to the OMM via its C-terminal transmembrane domain, with the N-terminus of Fis1 facing the cytosol [86]. Figure 4 depicts the structural elements of the Fis1 protein in greater detail. Overexpression of functional human Fis1 promotes mitochondrial fission, resulting in mitochondrial fragmentation [87]. Genetic and morphological studies have revealed that yeast Fis1 participates in the Drp1-mediated mitochondrial fission pathway [88]. Fis1 participates in mitochondrial fission functions by regulating the proper assembly of Drp1-containing complexes on the mitochondrial membrane and/or activating the fission activity of Drp1 complexes later in the process [87]. Overexpression of Fis1 caused mitochondrial fragmentation, which is compatible with these suggested roles of Fis1. Drp1 is more prevalent in the cytoplasm than in the mitochondria of mammalian cells in a steady state [89,90]. This suggests that Drp1’s association with mitochondria is transient and highly regulated [91].

According to structural analysis, the N-terminal region contains a tetratricopeptide repeat (TPR)-like core domain that differs from the typical TPR motif but is still important for protein binding [93]. Fis1 acts as a receptor in yeast for the recruitment of Dnm1p (Drp1 in mammals) to mitochondria via the adaptors mitochondrial division protein 1 (Mdv1) or CCR4-associated factor 4 (Caf4), thereby regulating mitochondrial morphology [94]. Since Fis1 is evolutionarily conserved from yeast to humans, mammalian Fis1 was initially thought to function similarly to its yeast ortholog Fis1, namely, as a receptor for the recruitment of Drp1 to mitochondria, thereby promoting fission. Several early studies found that overexpression of Fis1 produces substantial mitochondrial fragmentation and that knockdown of Fis1 increases mitochondrial elongation [95,96], indicating that Fis1 has a pro-fission role. Since mammals lack the yeast adaptors Mdv1 and Caf4, it is unclear whether Fis1 promotes mitochondrial fragmentation in a manner comparable to yeast. Depletion of Fis1, for example, has been shown to induce mitochondrial elongation in HeLa cells and in Fis1-null mouse embryonic fibroblasts (MEFs) [97,98], but has no effect on mitochondrial morphology in HCT116 (human colorectal carcinoma) cells [99]. Furthermore, increased or decreased Fis1 levels appear to have little effect on Drp1 subcellular distribution between the cytosol and mitochondria in mammalian cells. Increased levels of Fis1 in 293T (human embryonic kidney) cells, for example, have no effect on Drp1 subcellular distribution but promote mitochondrial fragmentation [100]. Similarly, knocking out Fis1 in HeLa and HCT116 cells does not reduce Drp1 levels on mitochondria [95]. In Fis1-null MEFs, however, Drp1 puncta on mitochondria are diminished to some extent. Taken together, these findings suggest that mammalian Fis1 is not required for Drp1 recruitment or Drp1-mediated mitochondrial fission but that it may play specific roles in certain physiological processes, cell types, or unique pathological pathways [101].

#### 2.1.3. Structural Elements and Roles of Mff in Mitochondrial Fission

Mff was discovered in *Drosophila melanogaster* cells through a small interfering RNA (siRNA) screen, and it exists in metazoans but not in yeast [102]. By means of alternative splicing, the Mff gene can produce at least nine different isoforms [95]. Mff, like Fis1, is also comprised of a C-terminal TM domain which anchors to the OMM and an N-terminal domain with three short amino acid repeats (R1-R3 motifs) and a coiled-coil domain that faces the cytosol. For the recruitment of Drp1 to OMM, the first 50 N-terminal residues containing R1 and R2 motifs are highly involved, and this is the minimal region mandatory for Drp1/Mff interaction (Figure 5) [103,104]. Mff depletion limits mitochondrial fission and generally prevents Drp1 recruitment to mitochondria in HeLa cells or MEFs [52], whereas Mff overexpression recruits the bulk of Drp1 from the cytosol to mitochondria and produces substantial mitochondrial fission in HeLa cells [35]. It is becoming clear that Mff is the primary receptor for Drp1 recruitment to mitochondria and, as a result, actively promotes mitochondrial fission in mammals [52].

#### 2.1.4. Structural Elements and Roles of Mid49 and Mid51 in Mitochondrial Fission

Mid49 and its paralog Mid51 (also known as MIEF2 and MIEF1, respectively) are important mitochondrial receptors for Drp1 recruitment to mitochondria in mammals [51]. Mid49 and Mid51 are mitochondrial proteins found only in vertebrates [105]. Overexpression of either Mid49 and Mid51, like the mitochondrial receptor Mff [35], causes extensive recruitment of cytosolic Drp1 to mitochondria in an Fis1- and Mff-independent manner [106]. Mid51 overexpression, on the other hand, causes mitochondrial elongation rather than fission in most cells [50]. Drp1 is thought to be inhibited by Mid49 and Mid51, which sequester it on the mitochondrial surface and reduce its GTPase activity [51,105,106].

Mid49 and Mid51 share 45 percent amino acid sequence similarity in humans, and both have an N-terminal TM domain that anchors them in the OMM, whereas Fis1 and Mff have a C-terminal TM domain (Figure 6) [105]. In addition, Mid49 and Mid51 can also form homodimers and heterodimers [105]. However, Mid49 and Mid51 differ in some ways. For example, biochemical analysis reveals that, in addition to the monomeric form, Mid51 appears primarily as dimers, whereas Mid49 appears as oligomers. Importantly, oligomerization of Mid49 requires the first 1–49 residues, including the TM domain, whereas dimerization of Mid51 requires the region between residues 109 and 154 but not the TM domain [105]. Furthermore, the distinct domains found in the crystal structures of Mid49 (PDB: 5WP9 [107]) and Mid51 (PDB: 5X9C [108]) have different functionalities. Mid51 can bind to nucleotide diphosphates (ADP and GDP), whereas Mid49 does not, and Mid51 binding to ADP can induce Drp1 oligomerization, self-assembly, and GTPase activity [49,108,109]. Interestingly, treatment of Mid49- or Mid51-overexpressing cells with antimycin A, an inhibitor of complex III of the electron transport chain, causes mitochondrial fragmentation in Mid51 but not in Mid49. Furthermore, this process also demands the binding of ADP to Mid51 [109]. According to a novel cryo-EM structural study, Drp1 has four interfaces that mediate interaction with Mid49 and Mid51, indicating strong structural evidence for Drp1 recruitment to mitochondria (PDB ID: 3LJB) [107,110].

## 3. New Insights into Mitochondrial Fission Protein Interaction Networks

The ever-growing complexity of mitochondrial fission protein interaction networks is orchestrated predominantly by evolutionarily conserved GTPases, with Drp1 co-assisted by a series of adaptor proteins, Fis1, Mff, Mid49, and Mid51. The mitochondrial dynamics highly rely on this interactome for communications and PPIs, which ensure cell-specific mitochondrial morphology, function, and intracellular distribution, as well as timely satisfaction of altering metabolic and energetic demands of cells. The information coordinating mitochondrial interaction hubs associated with fission and fusion proteins seems critical for revealing the underlying secrets behind the physiological and pathological progression of varied diseases, including neurogenerative disease, cancer, and CVDs. Drp1-mediated fission also allows caspase activation, cytochrome c release, and OMM permeabilization. However, the entire picture correlating the involvement of adaptor proteins still lacks clarity. Hence, exploring new insights into the fission protein-mediated interactome may allow the discovery of the missing links in fission events. In order to explore the updated fission-mediated PPI network involved in mitochondrial dynamics, we used the online STRING database platform to generate an updated representation of the fission interactome [111]. This PPI network (Figure 7) may shed light on the networking pattern of fission proteins and provide clarity about how Drp1 is connected to adaptor proteins in the dynamic fission process. Here, we explain the current evidence for Drp1-mediated fission protein networking and its regulatory role in fission and mitochondrial dynamics.

### 3.1. Role of Distribution and Oligomerization of Drp1 in PPI Networking

Drp1s are versatile dynamin fission-regulating proteins, controlling mitochondrial fission networking by mediating oligomerization, enabling spiral compaction and GTP-dependent conformational changes, facilitating mitochondrial membrane constriction. We have already discussed the multi-step process involved in Drp1-dependent mitochondrial fission and the various PTMs achieved by Drp1 that are mandatory for OMM targeting. The studies performed on the subcellular distribution of Drp1 provide an intuition that the entire mitochondrial Drp1 pool exists as a broad distribution of oligomers, with fission mainly relying on the quantitative distribution of cellular Drp1 [112]. The in vitro studies clearly mentioned that Drp1 maintains a pool of dimer–tetramer equilibrium in the cytosol. The specificity or affinity of Drp1 to bind the adaptor fission proteins and OMM receptors to fragment the mitochondrial network is mainly regulated by Drp1 cellular distribution, the oligomer pool, the presence of Drp1 isoforms, PTMs and GTPase activity. Mff favorably interacts with dimeric Drp1 rather than the monomer form and facilitates GTPase activity followed by Mid49- and Mid51-mediated Drp1 self-assembly/oligomerization [113]. Sorting and assembly machinery (SAM-50), a part of the SAM complex which inserts beta-barrels into the OMM, also showed interaction with Drp1 by monitoring outer and inner membrane fission and inducing Drp1-dependent mitochondrial fission through an unknown machinery [114]. Even though the canonical Drp1 receptors Mff, Mid49, and Mid51 may collaborate and promote fission, the details about fission complex formation remain vague.

### 3.2. Lack of Membrane-Anchored Domains in Drp1 Facilitates its PPI Networking

The recruitment of Drp1 from the cytosol to the mitochondrial surface is mediated mainly by four mitochondrial outer membrane anchored proteins: Fis1, Mff, Mid49, and Mid51. However, it was evidenced in previous studies that Fis1 interaction is not mandatory for the translocation of Drp1; for example, neither overexpression nor RNA interference (RNAi) knockout of Fis1 influenced the complete Drp1 distribution pattern of mitochondria [115]. Furthermore, the involvement of Fis1 and Mff proteins in the recruitment of Drp1 has been demonstrated, indicating that both of these proteins have independent roles in mitochondrial fission [52]. Remarkably, overexpression of cellular Fis1 strongly promotes mitochondrial fission, resulting in an accumulation of fragmented mitochondria. The structurally C-terminal tail of Fis1 is essential for mitochondrial localization, whereas the N-terminal region plays a pivotal role in mitochondrial fission [116].

### 3.3. Role of Mff in GTPase Activity of Drp1 and Oligomerization

Mff, being the key receptor protein, plays a very crucial role in Drp1 GTPase activation and oligomerization. Several studies support the oligomeric existence of Mff [48,117]. The oligomerization prediction studies performed on Mff by velocity analytical ultracentrifugation and size exclusion chromatography suggest the probability of the existence of two oligomeric forms, mainly a dimer or a tetramer [104,117,118]. Even though the accurate determination of Mff oligomerization stoichiometry demands further structural validation, the recent reports by Liu et al. suggest that cytoplasmic portion of Mff is oligomeric in nature and that this oligomerization is mandatory for the (i) recruitment of Drp1, (ii) Drp1 GTP hydrolysis, and (iii) punctual assembly of Drp1 on mitochondria, followed by (iv) peroxisomal and mitochondrial division [104,119]. The VD of Drp1 inhibits the aberrant interaction of Drp1 dimers with Mff, whereas a high concentration of Mff can trigger Drp1 oligomerization in a VD-independent manner. The role of Mff/actin synergy favors the preassembly of Drp1 oligomers and the Mff interactome network. The inhibitory role of VD in Drp1 oligomerization was found to be overcome by the binding of Drp1 with actin filaments in such a way that it can bind Mff more effectively [104,119]. The actin filament-mediated Drp1 oligomerization indirectly enhances Mff oligomerization by offering multi-valent binding sites for PPI networking and allows Mff to stimulate Drp1 assembly to trigger mitochondrial fission. The overexpression of Mff stimulates the recruitment of Drp1 to facilitate mitochondrial fragmentation, whereas the knockdown of Mff results in mitochondrial elongation and reduces recruitment of Drp1 to mitochondria [120].

### 3.4. Role of Actin Filaments in Drp1 Recruitment

Mff and actin filaments co-assist the journey of Drp1 activation in an oligomerization-dependent manner. The capability of Mff to assemble on mitochondria mainly relies on its oligomerization properties, as well as on Drp1 and actin filament binding properties [104]. Secondly, actin filaments play a pivotal role in fission by directly binding with Drp1, which enables Drp1 assembly and oligomerization. Then, Mff performs its role to recruit the actin-bound Drp1 oligomers to assemble the functional constrictive ring that facilitates the fission event [121]. Mff does not exhibit direct interaction with actin, but actin filaments synergize with oligomeric Mff to stimulate Drp1 nucleation and oligomerization to allow fission to proceed. Even though the influence of Mff on Drp1 in enabling the fission event is greater than that of actin, the complete picture correlating the role of Mff, actin filaments, and calcium effect in Drp1 recruitment/nucleation remains vague.

### 3.5. Role of Drp1 Phosphorylation Status in Mitochondrial Fission-Mediated Interactome

The phosphorylation of Drp1 is a well-explored PTM which has a central role in modulating Drp1 recruitment, adaptor protein interactions, and fission activity. Previous studies have pointed out that phosphorylation of Drp1 at Ser616 mediated by CDK1/cyclin B induces mitochondrial fission, whereas phosphorylation at Ser637 by the PKA inhibits the translocation of Drp1 from cytosol by impairing GTPase activity [122]. The dephosphorylation of Ser637 by calcineurin, on the other hand, promotes fission by triggering mitochondrial Drp1 translocation [122,123]. Even though the overall picture of the Drp1 phosphorylation-mediated fission interactome is still not clear, Drp1 phosphorylation status determines PPI networking with Mid49, Mid51, and Mff for fission. Interestingly, a recent report by Yu et al. suggested that phosphorylation of Drp1 at Ser637 is not a key controlling factor in the mitochondrial recruitment of Drp1 and Drp1-mediated peroxisomal fission [100]. They also showed the existence of phosphorylated Drp1 at Ser637 in both cytosol and mitochondria, where it interacts with Mid49, Mid51, and Mff. The knockdown of PKA arrests the phosphorylation of Drp1 at Ser637 and hence proves that the phosphorylation of Drp1 is not mandatory for Drp1/Mff interaction but enhances the interaction of Drp1 with Mid49 and Mid51 [100]. The overexpression of Mid49, Mid51, and Mff comparatively increased total Drp1 levels in mitochondria, indicating clear-cut roles for Mid49, Mid51, and Mff in Drp1 translocation [100]. Ultimately, the entire study concluded that Mid49 and Mid51 act as molecular bridges connecting Drp1 and Mff to form Drp1/Mff/Mids complexes in mitochondria and facilitate the direct binding of Drp1 to Mff, resulting in mitochondrial fragmentation.

### 3.6. Role of Fis1 in Drp1-Mediated Fission Networking

The structural analysis of Fis1 revealed the presence of a C-terminal TM domain exposed to the cytosol and a cytosolic domain at the N-terminal forming six alpha helices (1α–6α) coordinating a TPR [92]. The Fis1 TPR domain is heavily involved in regulating the Fis1/Drp1 interaction, particularly the first alpha helix (1α), which is found in the N-terminal domain and is known to construct the PPI network coordinating fission events [92,124]. The research findings of Yu et al. demonstrated that the overexpression of Fis1 resulted in excess fission and apoptosis, whereas the knockdown of Fis1 resulted in cytochrome c release and progression of apoptosis [125]. Fis1 is considered the key rate-limiting factor in mitochondrial fission by controlling the cytosolic recruitment and assembly of Drp1 in fission processes. The N-terminal 1α helix found in the TPR domain of Fis1 acts as the regulatory element that stabilizes/destabilizes Fis1/Drp1-interaction-mediated fission [125]. The accurate conformation of Fis1-TPR motifs also seems indispensable for providing a Drp1 binding site for fission-associated PPI networking [125]. The initial recruitment of Drp1 is mediated by Fis1 to achieve the proper concentration of Drp1 for self-assembly and binding activity. Successively, there occurs the dissociation of the 1α helix found in the Fis1-TPR domain of Drp1 to trigger the direct binding of Drp1 to OMM, catalyzing the GTP-mediated fission event [125]. Even though the complete recycling of Drp1 and its further recharge with GTP to enable fission processes is facilitated by destabilization of the TPR-mediated Fis1/Drp1 interaction, the entire mechanism remains unclear. The overexpression of 1α helix deleted Fis1 resulted in cytochrome c release, mitochondrial swelling, arrest of the fission process, and further promoted apoptosis [125].

### 3.7. Mid51-Mediated Recruitment of Drp1

Mid51 binds to Drp1 and aids the recruitment of Drp1 to OMM. Fis1 and Mff are fission-promoting adaptor proteins which serve as receptors for the recruitment of Drp1 to mitochondria [115]. Zhao et al. experimentally proved that Mff and Fis1 are not mandatory for Mid51-mediated recruitment of Drp1 to mitochondria. These findings revealed that depletion of Fis1 alone does not make any alterations to Drp1 recruitment status, whereas silencing of Mff reduces mitochondrial localization of Drp1. Later when, Mid51 was ectopically expressed, a transient variation was observed in the Drp1 recruitment to mitochondria that was not mediated/influenced by silenced Fis1 and Mff [50]. The Mid51-mediated Drp1 recruitment is not dependent on the phosphorylation status of Drp1. Mid51 can effectively bind to both phosphorylated and non-phosphorylated (Ser637) Drp1 [50]. Recent studies also emphasize that Mid51 plays a central role in inhibiting Drp1-induced fission, whereas Mffs promote Drp1-induced fission and act as both positive (Mff) and negative (Mid51) regulators in controlling Drp1-mediated fission events [50,51]. The overexpression of Mid51 brings two mitochondrial membranes into close proximity and facilitates extensive mitochondrial fusion, while the depletion of Mid51 initiates mitochondrial fragmentation [51,105,106].

### 3.8. Role of Mid51 Interaction with Fis1 in Drp1-Induced Fission

Mid51 acts as a bridging protein in the Drp1-mediated interactome because it can interact with both Fis1 and Drp1 via its non-overlapping domains. The relative levels of Fis1 and Mid51 are considered to be the rate-limiting factors in the promotion of Drp1-mediated fission or fusion [50]. Elevated levels of Mid51 promote the Mid51/Drp1 interaction that further inhibits Drp1-mediated fission by disturbing GTPase activity. High levels of Fis1 encourage Mid51/Fis1 interaction, leading to Drp1-mediated mitochondrial fission events [50]. The studies performed have unveiled an innovative mechanism which remotely controls both mitochondrial fusion and fission machinery.

### 3.9. Mid49/Mid51-Orchestrated Drp1-Mediated Fission

Mid49 and Mid51 anchored in the OMM play central roles in balancing mitochondrial dynamics by participating in both fusion and fission events [105]. Recently, there have been several reports that Mid49 and Mid51 have opposing roles in Drp1 recruitment to mitochondria. Palmer et al. showed that the overexpression of Mid49 and Mid51 blocked fission by inactivation of Drp1, thereby leading to unopposed fusion events supported by fusion mediators Mfn1 and Mfn2. In addition, peroxisomal elongation was also observed during overexpression of Mid49 and Mid51 [101]. Even though the proposed functions of Mid49 and Mid51 in mitochondrial dynamics have been under debate, both Zhao et al. and Palmer et al. proposed that Mid49 and Mid51 serve as mitochondrial fission mediators which can act independently on Drp1 recruitment [101,126]. Mid51/Mid49 also interacts with Drp1 via Mff and Fis1 in an independent manner, but, interestingly, the knockdown of Mid51/Mid49 adversely results in mitochondrial elongation due to Drp1 inactivation, hence blocking fission [127].

### 3.10. Mff-, Fis1-, Mid49-, and Mid51-Mediated Drp1 PPI Networking in Fission Regulation

Currently, there remains controversy regarding the clarity and connectivity of the adaptor proteins (Mff, Fis1, Mid49, and Mid51) and their individual roles in the Drp1-mediated fission interactome. The Drp1-involved interplay of these adaptor proteins and mitochondrial connectivity was newly elucidated by gene editing technology. The studies were performed on mouse embryonic fibroblast cell lines to elucidate the PPI in the Drp1-involved fission mechanism [106]. The individual knockout studies of each adaptor protein—Fis1, Mff, Mid49, and Mid51—did not find any drastic alterations in mitochondrial morphology and Drp1 localization to mitochondria, whereas multiple deletions were adversely reflected in mitochondrial connectivity, Drp1 translocation, and association [106]. Mff deletion individually caused peroxisomal elongation, whereas additional deletion of Mid49, Mid51, and Fis1, along with Mff deletion, did not further enhance peroxisomal length. Hence, there is a strong indication that, out of the four core adaptor proteins, Mff contributes more to peroxisome division than Mid49, Mid51, and Fis1. Mid49, Mid51, and Mff are capable of recruiting a fission-competent pool of Drp1s independently, whereas Fis1 was found to be unable to recruit individually [106]. Previous reports suggest the close proximity of adaptor protein Mid51 and Mff [128]. The proximity-based biotin labeling (PBL) approach disclosed neighboring associations between Mid51, Mff, and Drp1 but not with Fis1, and the studies of their roles in stimulation of Drp1 GTPase activity disclosed the inevitable roles of Mid51 and Mff in regulating Drp1 GTPase activity in opposing directions [106]. We have already discussed the bona fide role of Fis1 as an adaptor protein and yet it still seems controversial. However, Osellame et al. recently concluded that Fis1 deletion did not induce morphological changes in mitochondria [129], and PBL studies revealed that Fis1 was not found to be close to Drp1, thus assigning it a profound role in stress-induced mitochondrial fragmentation [130]. Mff mutations cause mitochondrial-associated illnesses, such as encephalopathy, ocular atrophy, and neuromuscular defects [131], while a Mid49 mutation causes muscle myopathy [53]. In general, cytosolic Drp1 is recruited to the mitochondrial outer membrane and binds to receptor/adaptor proteins (such as Fis1, Mff, Mid49, and Mid51) and oligomerizes to form a ring-like structure that divides mitochondria by constriction [52].

## 4. Inhibition of Mitochondrial Fission Protein Interactions

Mitochondrial fission proteins are a specific type of proteins in the organism, which are associated with biological membranes and lipid domains. The ligands can target mitochondrial fission proteins to regulate their biological functions as drugs [132]. Understanding of the genomics, transcriptomics, structures, interactions, biological systems, and functional profiles of the therapeutic targets benefits the development and practice of modern medicine. Over the past decade, numerous freely accessible databases have provided comprehensive information about drugs, therapeutic targets, and drug-targeted pathway information for facilitating drug discovery efforts [133,134].

Targeting and inhibiting proteins involved in mitochondrial dynamics has gained immense popularity. Since mitochondrial fragmentation is a hallmark of many pathological processes, finding a way to inhibit excess mitochondrial fission may improve patient outcomes. Currently, mitochondrial division inhibitor 1 (mdivi-1) and P110 are two of the most thoroughly researched fission regulators. Mdivi-1 is a quinazolinone derivative small molecule inhibitor discovered through a chemical screen in yeast and is a proposed Drp1 inhibitor [135], whereas P110, a small peptide derived from a Fis1/Drp1 homology sequence, is thought to disrupt the Fis1/Drp1 PPI in a specific manner [136]. In recent years, short peptides functioning as effective inhibitors of Drp1 have also been developed to modulate mitochondrial dynamics, such as P110 and P259 [62,136]. Furthermore, Drp1 GTPase inhibition with mdivi-1 or disruption of the Drp1/Fis1 PPI with P110, leading to reduce mitochondrial fission, results in improved mitochondrial structure and function [137]. P259, a Drp1/Mff-specific PPI inhibitor, on the other hand, is limited due to its inhibitory effect on physiological fission, which may not be as protective as originally predicted and may actually hasten disease progression [138]. The mechanism regulating fission has emerged as a promising therapeutic approach due to the association between excessive mitochondrial fission and several diseases [62]. These findings suggest that physiological Drp1 activity is an important component of mitochondrial function [62,138]. Protecting physiological fission by inhibiting adaptor-specific interactions with short peptides such as P110 holds promise for clinical applications and drug development in the future. The most-researched intervention targets for inhibiting mitochondrial fission functions are summarized in Table 2.

Given Drp1-dependent mitochondrial fission’s multiple roles in the development and progression of CVDs, the expression and activity of Drp1 can be manipulated pharmacologically or genetically to prevent and treat cardiac dysfunction [62]. For this purpose, specific chemical compounds have been identified. Chemical screening, for example, revealed that mdivi-1 inhibits Drp1 function by repressing its GTPase activity. Treating Drp1 cells with mdivi-1 can limit ischemic stress and protect the heart against I/R injury by preserving mitochondrial morphology, improving mitochondrial function, preventing apoptosis, and inhibiting mitochondrial permeability transition pore (mPTP) opening [145]. Furthermore, mdivi-1 treatment can increase ATP levels as well as mitochondrial complex I, IV, and V activity in diabetic hearts. Furthermore, in diabetic mice, mdivi-1 reduced superoxide anion production, decreased cardiomyocyte apoptosis, and improved cardiac dysfunction [146]. In general, midivi-1 is an inhibitor of Drp1 and inhibits the permeabilization of the outer membrane [147]. However, the effect of mdivi-1 appears to be time- and dose-dependent because complete or long-term inhibition of Drp1 causes harmful effects in diabetic cardiomyopathy and cardiac hypertrophy by promoting the accumulation of damaged mitochondria and impairing cardiac function [73,139,148]. In vitro and in vivo, intervention with mdivi-1 or Drp1 siRNA can suppress excessive mitochondrial fission, maintain mitochondrial membrane potential, and inhibit cell death after I/R-induced brain injury [73,148]. Aside from Drp1 inhibition, mdivi-1 contributes to its protective role under I/R conditions by suppressing ROS production and preventing cytochrome c release [73]. Ginkgolide K is a lactone compound that is found in *Ginkgo biloba* leaves which reduces mitochondrial fission, resulting in decreased neuronal apoptosis in the I/R process by reducing mitochondrial Drp1 recruitment and increasing Drp1 phosphorylation at Ser637 [73,140]. Drp1 activity can be inhibited by recruiting PKA to the OMM to phosphorylate Drp1 at Ser637, and A-kinase anchoring protein 1 (AKAP1) protects neurons from I/R injury [73,144,149].

## 5. Mitochondrial Fission Proteins and Cardiovascular Diseases

Cardiovascular diseases (CVDs) are among the leading causes of mortality, and recent evidence has shown an unsurprising link connecting mitochondrial dysfunction and CVDs [150]. Mitochondria account for about 30–40% of cardiomyocyte volume; hence, it seems logical that cardiomyocyte function is closely relatable to healthy mitochondrial status/integrity [151]. Mitochondrial dynamics play a supervisory role by balancing fusion, fission, and mitophagy events to meet the bioenergetics of major consumer cells, such as cardiac, skeletal muscle, and brain cells. A healthy mitochondrial network is highly mandatory for proper cardiovascular, neurodegenerative, and endocrine functioning, so mitochondria are central players that are receiving ever greater attention. The increasing evidence validates that the genetic and pharmacological targeting of mitochondria can ameliorate pathological conditions in their early stages by offering potential therapies [150,151,152]. During mitochondrial damage, fusion and fission constitute a harmonizing response to maintain a network of healthy mitochondria. The heart, being the body’s highest energy consumer, relies predominantly on mitochondria to meet its massive metabolic energy demands due to its rhythmic excitation and contraction cycles [153]. Mitochondrial dysfunction has been associated with varied mutations in electron transport chain subunits, mtDNA transcription and translation machinery, mitochondrial tRNAs, and mitochondrial dynamics proteins [154]. Mitochondrial dysfunction generates ROS, leading to oxidative stress, apoptosis, and cell death. Vascular endothelial cells are highly sensitive to these oxidative stresses, and they elicit immune reactions that damage the vascular wall, resulting in endothelial dysfunction and atherosclerotic plaque formation [62,151]. Hence, disruptions to mitochondrial homeostasis adversely affect cardiomyocyte survival rates and lead to the progression/development of CVDs, such as atherosclerosis, myocardial injury, endothelial dysfunction, cardiac hypertrophy, heart failure, I/R injury, and myocardial infraction (Figure 8) [155]. The functional impairment of cardiac mitochondria in CVDs may lead to deprived ATP production, increased cell apoptosis, enhanced oxidative stress, and impaired autophagy.

We have also highlighted the updates on Drp1-mediated fission networking in mitochondrial dynamics and their importance in maintaining mitochondrial quality control in the heart. Repressed Drp1 expression induces mitochondrial depolarization in the heart under both basal and stressful conditions [156]. According to preliminary experimental findings on mitochondrial dynamics in the context of myocardial I/R injury, inhibiting Drp1-mediated mitochondrial fission triggered by myocardial ischemia may be a therapeutic strategy for shielding the heart against acute I/R injury (Figure 8). The interplay between Drp1-mediated or other mitochondrial fission proteins (Fis1, Mff, Mid49, or Mid51) mediated disturbances in mitochondrial homeostasis which may lead to a series of complex CVDs, such as myocardial I/R injury, heart failure, and endothelial dysfunction [62,157]. The most-investigated proteins involved in mitochondrial fission are Drp1 and Fis1 [158]. However, mutations in Mff also lead to mitochondrial disease (including encephalopathy, optic atrophy, and neuromuscular defects) [131], and a mutation in Mid49 leads to muscle myopathy [53]. These findings suggest that alterations in mitochondrial shape are essential to sustain the metabolic needs of cardiac differentiation, ranging from mitochondrial fragmentation in undifferentiated stem cells to a more elongated and interconnected mitochondrial network [159]. The evidence accumulated in the literature has established the pivotal role of mitochondrial dynamics and the CVDs associated with them. Thus, in this review, we have summarized the updated knowledge about the relationships between mitochondrial fission proteins, focusing mainly on Drp1 and its influence on CVD pathogenesis, with an emphasis on the different CVDs described below.

### 5.1. Atherosclerosis

Atherosclerosis is one among many chronic inflammatory conditions in arteries leading to lipid accumulation, endothelial injury, vascular inflammation, abnormal lipid metabolism, and mitochondrial oxidative stress. During the initial stages of atherosclerosis, lipid accumulation promotes plaque formation and, along with disturbed mitochondrial ROS production levels, induces endothelial dysfunction (ED) [160]. ED further accelerates the accumulation of oxidized low density lipoproteins (ox-LDLs) in the walls of the arteries, further inducing mPTP formation, calcium influx, and, ultimately, mitochondrial damage [161]. ROS production simultaneously regulates both Drp1-mediated fission and fusion events in mitochondria. High blood glucose levels excite Drp1 activity in endothelial cell mitochondria, leading to more fission and ROS production as a by-products [162]. The known Drp1 inhibitor mdivi-1 reduced the aortic cell injury and oxidative stress induced by excessive fission. In many cases, Drp1 induced disturbances in mitochondrial dynamics that promoted the depolarization of mitochondria in cardiac cells. In vivo experimental studies proved that mitochondrial DNA damage reduced mitochondrial respiration and ATP contents, exacerbating atherosclerotic conditions. Further therapies, such as proliferation, migration, and vascular remodeling, are mandatory to rupture atherosclerotic plaques [163]. Drp1 is highly involved in the calcification process of arteries during atherosclerosis progression, while the osteogenic differentiation of human vascular smooth muscle cells upregulated Drp1 expression [164]. Both in vitro and in vivo experiments proved that mdivi-1-mediated inhibition of Drp1 overexpression during osteogenic differentiation attenuates mitochondrial dysfunction, matrix mineralization, and alkaline phosphatase activity and arrests atherosclerosis progression by attenuating the calcification process of arteries [164].

### 5.2. Cardiac Hypertrophy

Cardiac hypertrophy (CH) is an early adaptive response characterized by enhanced myocardial contractility, increased myocardial cell volume/mass, and greater ventricular wall thickening. Additional sarcomeres are synthesized to compensate for cardiac workload/hypertension in cardiomyocytes [165]. It has been evidenced that during CH mitochondrial fission is hyperactivated and that, mechanistically, the pathogenesis of CH is greatly affected by mitochondrial metabolism and total mitochondrial dynamics. Physiological hypertrophy develops as an adaptive response during exercise by enhancing Drp1-mediated mitochondrial fission to facilitate mitochondrial clearance by altering bioenergetics [166]. The pathological stress induced by hypertrophy ultimately results in reduced ATP production by mitochondria, loss of cardiomyocytes, cardiac remodeling followed by diastolic dysfunction, and, finally, heart failure [167].

The experimental analysis of the cardiac hypertrophy pathway in phenylephrine-treated rat neonatal cardiomyocytes and transverse-aortic-banding-treated mouse hearts indicated the sequential process of mitochondrial translocation of Drp1, Drp1 phosphorylation, and Drp1-mediated mitophagy [168]. The Drp1-mediated hypertrophy pathway enhanced mitochondrial numbers due to excessive fission and resulted in loss of mitochondrial function. Mdivi-1 was found to be prevalent in the treatment of CH by suppressing ROS production by inhibiting the activity of Ca^2+^-activated protein phosphatase calcineurin and calmodulin-dependent protein kinase II [169]. During cardiac hypertrophy, besides excessive fission, Drp1-mediated mitophagy also plays a critical role in the self-regulation of cardiomyocyte function and survival. Intact mitophagy was activated in mouse hearts during CH for the proper regulation of oxidative stress, protection of vascular endothelia, and mitochondria-associated cell death. It was accompanied with phosphorylation of Drp1 at Ser616 and promoted fission by triggering mitochondrial Drp1 translocation [170]. Moreover, studies performed on cardiac-specific heterozygous Drp1-knockout mice demonstrated that Drp1 inhibition repressed B-cell lymphoma 2 (BCL2) interacting protein 3 (Bnip3)-induced mitophagy and led to the accumulation of damaged mitochondria in the heart [170]. All these findings clearly point to Drp1-mediated fission and mitophagy being associated with cardiac hypertrophy.

### 5.3. Myocardial Infarction (MI)

MI is defined as the occlusion of a coronary artery that leads to cardiac ischemia and successive impairment of heart ventricular function by abolishing hemodynamic stability [171]. It has been reported that Drp1-driven fission contributes to myocardial cell death during pressure overload due to occlusions in MI. The clinical/therapeutic approach to inhibiting Drp1 GTPase activity by using mdivi-1 or P110 reduced infract size and attenuated mPTP via the refinement of cardiovascular functioning in MI animal models [172,173]. siRNA-mediated Drp1 inhibition also improved cardiomyocyte contractility by regulating cytosolic potassium and calcium levels [172].

### 5.4. Myocardial Ischemic/Reperfusion (I/R) Injury

Myocardial infarction can result from the rupturing of atherosclerotic plaque in the coronary artery. In order to salvage the ischemia condition, myocardium reperfusion must be performed to re-establish heart blood and oxygen delivery [174]. Paradoxically, mitochondrial fission is activated initially during ischemia and sustained during reperfusion. This may also be accompanied by some drastic changes in mPTP, production of ROS, and variation in intracellular calcium and sodium distribution in cardiomyocytes. Thus, coronary occlusion and successive reperfusion itself can cause cardiomyocyte death and irreversible myocardial injury, termed myocardial I/R injury [175,176]. Drp1 activation was experimentally shown in peri-infarct regions of mouse hearts and was manifested in noticeable cardiac dysfunction and mitochondrial fission [177]. Excessive fission disrupts heart integrity by affecting endothelial barrier function, whereas restricting fission in cardiac endothelial cells reverses mitochondrial damage and cell death in the case of induced I/R injury. The pharmacological inhibition of Drp1-mediated fission by mdivi-1 in myocardial I/R injury diminishes mPTP formation and cell death in cardiomyocytes and there by prevents long-term cardiac dysfunction by inhibiting fission at the onset of reperfusion [176].

There is a plethora of biological consequences if mitochondrial fission/fusion is disrupted, and these have an impact on a range of functions, including ER interactions, Ca^2+^ signaling, autophagy, and apoptosis. In summary, these data demonstrate that fusion and fission processes coordinately sustain cardiac function and suggest preservation of the balance of fission and fusion as a promising therapeutic strategy for cardiac disease.

## 6. Conclusions

The entire balance between the multifaceted process of mitochondrial fission and fusion plays a critical role in regulating mitochondrial dynamics and cardiac homeostasis. In this review, we have summarized the current research reports on Drp1-dependent fission and the Drp1-mediated mitochondrial fission protein interactome and their significance in regulating mitochondrial dynamics and cardiac pathology. The GTPase activity of Drp1 and its interactions with specific adaptor proteins—Fis1, Mff, Mid49, and Mid51—exist as indispensable regulators of fission events coordinating mitochondrial dynamics. PTMs induce complexity and variation that can alter protein stability and function. After translation, proteins can be modified by phosphorylation, SUMOylation, ubiquitination, and S-nitrosylation. A better understanding of the structural elements, interaction networks, and roles of all these fission proteins will provide valid clues to discover the missing links in fission events. Intact mitochondrial dynamics appear to be mandatory for the maintenance of normal cardiovascular function to maintain the high energy demands of cardiomyocytes. Drp1-driven fission in turn influences physiological and pathophysiological conditions within cells. The in vivo experimental data indicate that acute inhibition of mitochondrial fission events can reduce MI size and preserve cardiac homeostasis, whereas the chronic inhibition of fission seems to be detrimental because it suppresses mitophagy. Therefore, targeting Drp1-mediated fission is a double-edged sword, since balanced fission is necessary to maintain mitochondrial homeostasis and physiological mitochondrial fission under stress seems mandatory to meet high energy demands. Thus, chronic inhibition of mitochondrial fission under CVDs also seems adverse. However, further clarification at several levels, such as the molecular and genomic levels, and of the mechanisms of action in mitochondrial dynamics are needed to evaluate the therapeutic approaches to fission machinery at the pharmacological level, which may lead to clinical translations and drug discovery in the future. In conclusion, exploring the new insights about Drp1-mediated fission interactions and the mechanisms regulating PPI may help to forge key therapeutic pharmacological links supporting clinical translations to reduce morbidity and mortality from CVDs.

## Figures and Tables

**Figure 1 ijms-24-05785-f001:**
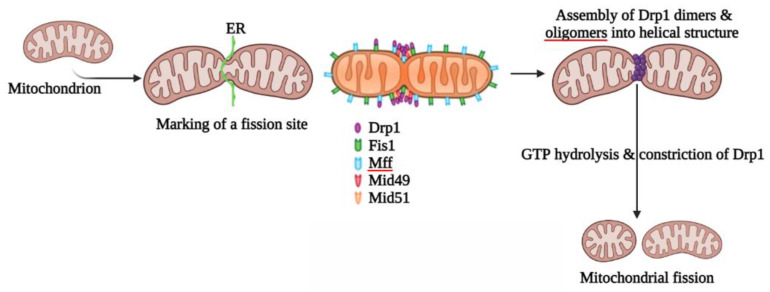
Mechanisms of mitochondrial fission process.

**Figure 2 ijms-24-05785-f002:**
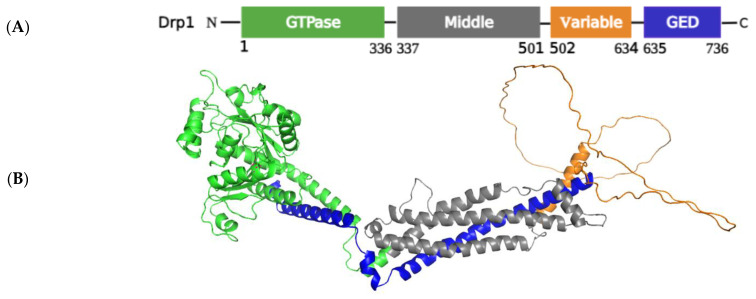
Architecture of Drp1 (isoform 1, full-length human Drp1 (736 aa)). (**A**) Drp1 has an N-terminal GTPase effector domain (GTPase, green), middle domain (gray), variable domain (VD, orange), and C-terminal GTPase effector domain (GED, blue). (**B**) Three-dimensional structure of the Drp1 protein shown in colors that correspond to the respective domain colors (PDB: 3W6O [59]). PyMol (Schrodinger LLC) was used to generate the figure [64].

**Figure 3 ijms-24-05785-f003:**
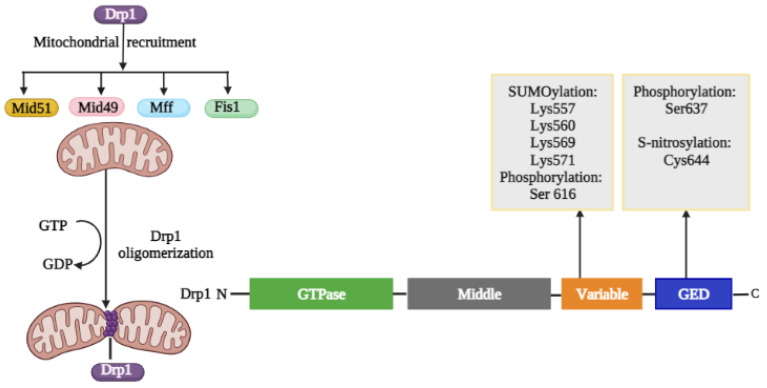
A schematic of the activation of fission linkage with Drp1 PTMs. The adaptor proteins Fis1, Mff, Mid49, and Mid51 recruit cytosolic Drp1 monomers to the outer mitochondrial membrane, facilitating the formation of helical ring Drp1 oligomers. Drp1 stimulates OMM constriction and scission by hydrolyzing GTP. Drp1’s VD and GED contain various residues that can be modified by different types of PTMs that regulate mitochondrial fission.

**Figure 4 ijms-24-05785-f004:**
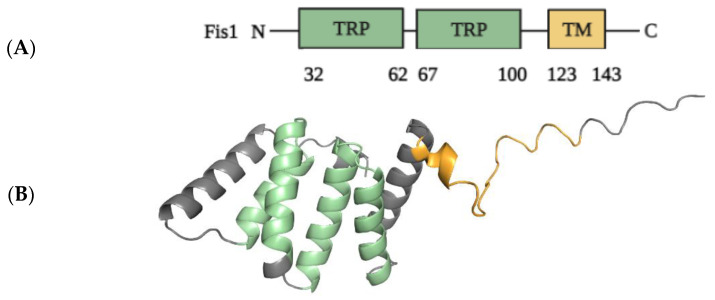
Schematic representation of the structural elements of Fis1 protein (152 AAs). (**A**) Fis1 has two core tetratricopeptide repeat (TPR, green) structure motifs flanked by two helices (helices 1 and 6). The N-terminal region of Fis1 is referred to as the “N-terminal arm”. The transmembrane domain (TM, yellow) structure in the membrane is unknown but presumably helical. (**B**) Three-dimensional structure of Fis1 protein shown in colors that correspond to the respective domain colors (AlphaFold: AF-Q9Y3D6-F1) [92]). PyMol (Schrodinger LLC) was used to generate the figure [64].

**Figure 5 ijms-24-05785-f005:**
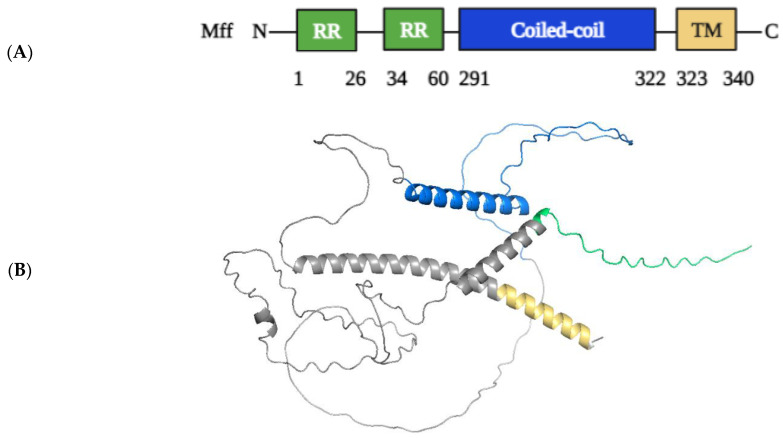
Schematic representation of the structural elements of Mff proteins (342 AAs). (**A**) Mffs have two N-terminal repeat regions domains (RR, green), a coiled-coil domain (blue), and a C-terminal transmembrane domain (TM, yellow). (**B**) Three-dimensional structure of Mff protein shown in colors that correspond to the respective domain colors (AlphaFold: AF-Q9GZY8-F1 [35]). PyMol (Schrodinger LLC) was used to generate the figure [64].

**Figure 6 ijms-24-05785-f006:**
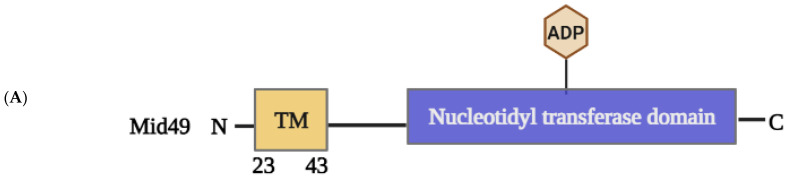
Schematic representation of the structural elements of Mid49 and Mid51 proteins. (**A**) Mid49 has an N-terminal transmembrane domain (TM, yellow) and a nucleotidyl transferase domain (ADP, blue). (**B**) Three-dimensional structure of the Mid49 protein shown in colors that correspond to the respective domain colors (PDB: 5WP9 [107]). (**C**) Mid51 has an N-terminal transmembrane domain (TM, yellow), Drp1 binding domain (Drp1 binding, magenta), and nucleotidyl transferase domain (ADP, blue). (**D**) Three-dimensional structure of the Mid51 protein shown in colors that correspond to the respective domain colors (PDB: 5X9C [108]). PyMol (Schrodinger LLC) was used to generate the figure [64].

**Figure 7 ijms-24-05785-f007:**
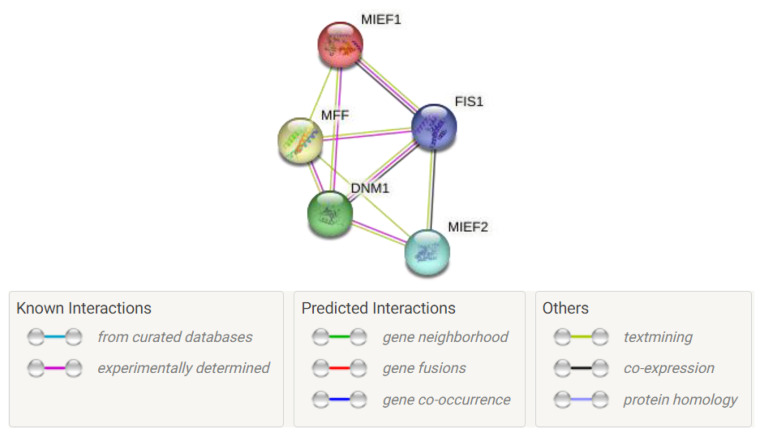
The protein–protein interaction network of mitochondrial fission and dynamics-related proteins of Drp1 (DNM1). PPI networks: dynamics-related proteins of Drp1 (DNM1) (lime-green node), mitochondrial fission protein (Fis1) (blue node), mitochondrial fission factor (Mff) (yellow node), mitochondrial dynamics 49 (Mid49/MIEF2) (gray node), and mitochondrial dynamics 51 (Mid51/MIEF1) (red node).

**Figure 8 ijms-24-05785-f008:**
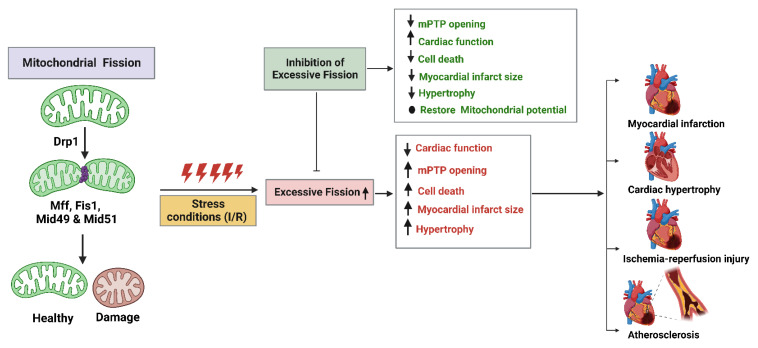
Involvement of mitochondrial fission proteins in cardiovascular diseases (CVDs). The fission process is primarily carried out by dynamin-related protein 1 (Drp1) and mitochondrial outer membrane mitochondrial fission 1 (Fis1), mitochondrial fission factor (Mff), mitochondrial dynamics 49 (Mid49), and mitochondrial dynamics 51 (Mid51). Under pathological stress conditions, excessive mitochondrial fission and imbalance in mitochondrial quality-control mechanisms occur. This may result in cytochrome C increase, mitochondrial permeability transition pore (mPTP) opening, and ATP decrease in several cardiovascular diseases (CVDs), including myocardial infarction, cardiac hypertrophy, ischemia/reperfusion (I/R) injury, and atherosclerosis. Pharmacological inhibition of excessive fission during I/R increases cardiac output by restoring mitochondrial membrane potential and reduces mPTP opening, infract size, and cell death. The figure was created with BioRender.com.

**Table 1 ijms-24-05785-t001:** List of the major mitochondrial fission proteins.

Name of Protein	Gene Name	Yeast Ortholog	Uniport ID	Human Homolog	Location	Molecular Functions	References
Dynamin 1	DNM1	DNM1, YLL001W, L1381	P54861	Drp1 or DNM1	Cytoplasm and cytosol	GTPase activity, GTP binding, identical protein binding, microtubule binding, protein kinase binding, RNA binding	[29,30,31]
Dynamin 1-like protein	DNM1L	Dnm1p	O00429	DNM1L, DLP1, DRP1	Cytoplasm and cytosol	GTPase activator activity, GTPase activity, GTP binding, GTP-dependent protein binding, identical protein binding, lipid binding, microtubule binding, protein homodimerization activity, small GTPase binding, ubiquitin protein ligase binding	[32,33]
Dynamin 2	DNM2	Dnm2	P50570	DNM2, DYN2	Cytoplasm and cytosol	D2 dopamine receptor binding, enzyme binding, GTPase activity, GTP binding, microtubule binding, nitric-oxide synthase binding, phosphatidylinositol 3-kinase regulatory subunit binding, protein-containing complex binding, protein kinase binding, SH3 domain binding, WW domain binding	[30,31]
Dynamin 3	DNM3	Dnm3	Q9UQ16	Drp3 or DNM3	Cytoplasm and cytosol	GTPase activity, GTP binding, identical protein binding, microtubule binding, nitric-oxide synthase binding, structural constituent of postsynapse, type 1 metabotropic glutamate receptor binding, type 5 metabotropic glutamate receptor binding	[30,34]
Mitochondrial fission 1 protein	FIS1	Fis1, MDV2, YIL065C	Q9Y3D6	Fis1, TTC11, CGI-135	OMM and peroxisome membrane	Identical protein binding	[35,36]
Ganglioside-induced differentiation-associated protein 1	GDAP1	-	Q8TB36	GDAP1	OMM	Regulates the mitochondrial network by promoting mitochondrial fission	[37,38]
Putative gametogenetin-binding protein 1	GGNBP1	-	Q5YKI7	GGNBP1	Cytoplasm and cytosol	Involved in spermatogenesis	[39]
Vascular endothelial growth factor receptor 2	KDR	-	P35968	KDR, FLK1, VEGFR2	ER and plasma membrane	ATP binding, cadherin binding, growth factor binding, heat shock protein 90 (HSP90) binding, identical protein binding, integrin binding, protein serine/threonine/tyrosine kinase activity, transmembrane receptor protein tyrosine kinase activity, vascular endothelial growth factor-activated receptor activity, vascular endothelial growth factor binding	[40,41,42,43,44]
Calcium uniporter protein, mitochondrial	MCU	-	Q8NE86	MCU, C10orf42, CCDC109A	Inner mitochondrial membrane (IMM)	Calcium channel activity, identical protein binding, uniporter activity	[45,46]
Mitochondrial fission factor	MFF	-	Q9GZY8	Mff, C2orf33, AD030, AD033, GL004	OMM and peroxisome	Identical protein binding, protein homodimerization activity	[47,48]
Mitochondrial dynamics protein MiD51	MIEF1	-	Q9NQG6	MIEF1, Mid51, SMCR7L	OMM	ADP binding, GDP binding, identical protein binding	[49,50]
Mitochondrial dynamics protein MiD49	MIEF2	-	Q96C03	MIEF2, Mid49, SMCR7	OMM	Regulation of mitochondrial organization, recruitment and association of the fission mediator dynamin-related protein 1 (DNM1L)	[51,52,53]
Mitochondrial fission process protein 1	MTFP1	-	Q9UDX5	MTFP1, MTP18, HSPC242, My022	IMM	Apoptotic process, mitochondrial fission, response to muscle activity	[54,55]
Mitochondrial division protein 1	Mdv1p and Caf4p	MDV1, FIS2, GAG3, NET2, YJL112W, J0802	P47025	-	OMM	Ubiquitin binding	[56]

**Table 2 ijms-24-05785-t002:** Inhibition of mitochondrial fission functions.

Group	Inhibitors	Mechanisms	Function	References
Peptide	P110	Inhibiting Fis1/Drp1 PPI, enzyme activity and Fis1 TPR-binding peptide	Inhibition of mitochondrial fission	[136]
	P259	Drp1/Mff PPI specific inhibitor	Inhibitory effect on physiological fission	[138]
Small molecule	Mdivi-1	Inhibiting Drp1 activity	Inhibition of mitochondrial fission	[139]
	Ginkgolide K	Reducing Drp1 recruitment	Inhibition of mitochondrial fission	[73,140]
siRNA	Drp1 siRNA	Suppressing of Drp1 expression	Inhibition of mitochondrial fission	[141]
	PPAR agonist	Supporting of Drp1 phosphorylation at Ser616	Inhibition of mitochondrial fission	[142,143]
Protein	AKAP1	Increase the Drp1 phosphorylation at Ser637	Inhibition of mitochondrial fission	[144]

P110 and P259: inhibiting peptides; Mdivi-1: small molecule Drp1 inhibitor; AKAP1: A-kinase anchoring protein 1; PPAR: peroxisome proliferator-activated receptor.

## Data Availability

Not applicable.

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
