# Peer review of "The Drp1-Mediated Mitochondrial Fission Protein Interactome as an Emerging Core Player in Mitochondrial Dynamics and Cardiovascular Disease Therapy"

_ijms, 2023, doi:10.3390/ijms24065785_

Round 1
Reviewer 1 Report
This manuscript aims to discuss the role of Drp1 in mitochondrial dynamics and cardiovascular disease therapy. The authors started from introducing mitochondria fission mechanism, followed by fission proteins and interaction networks. The manuscript then focused on the inhibitors of mitochondrial fission. In the end, the authors discussed the mitochondria fission proteins and cardiovascular diseases. The information presented in the manuscript will be helpful for the researchers that are new to the field.
Concerns and suggestions
1. Although the authors collected references from more then 160 papers. However, some critical references are not cited properly. For example, Ref. 111 and 118 for Line 512 may not be the best references. So as Ref. 27, 61 and 65. There are more in the manuscript.
2. Figure 8 did not fit the purpose to demonstrate the involvement of mitochondrial dynamics¸ quality control and cardiovascular diseases. The information is fragmented and have no connection. It is the reviewer’s suggestion that the author to remake to the figure to present the most critical concept of the manuscript.
3. The manuscript used lot of compound and complex sentences to descript multiple ideals. Some of them are confusing. It is highly recommended to break the sentence so that readers will be able to catch up the concept.
Author Response
See attached letter

Reviewer 2 Report
This review article is focused on Drp1 protein translocation and oligomerization, associated adaptor proteins, and post-translational modifications, as the principal machinery that underlies the triggering of mitochondrial fission to isolate and eliminate mitochondrial fragments that are no longer functional. The review focuses on cardiomyocyte mitochondria, which are relatively immobile, and the role of cell and mitochondrial stress that leads to fission, quality control and mitophagy.
Overall, the description of the proteins involved in fission, understood by gene knockout or protein suppression, is comprehensive and aided by useful diagrams and tables. Fission is described as a normal and necessary process to remove damaged mitochondria and encourage emergence of healthy mitochondria. This could occur in normally functioning cells but become accelerated in cells damaged by oxidative stress (oxygen, substrate deprivation leading to ROS production, reduced membrane potential and ATP production and mitochondrial swelling. (MPT pore opening is more likely associated with mitochondrial rupture and not a cause or result of fission; but fission may obviate MPT pore opening.) The brief summary at the end of each section is helpful to understand overall impact.
The weakest aspects of the review are the durth of information on the timing sequence and cause-effect relationships of Drp1 translocation to the OMM, the triggers in dysfunctional mitochondria that activate Drp1, the contradictory information on the roles or necessity of the associated proteins and PTMs in triggering fission, and the loose association on the precise role and overall contribution of fission in cardiovascular diseases. For example, normal fission is required for quality control, so if one either induces or inhibits fission by genetic or drug means, the result could be bad for the mitochondriome as well as for the cells the mitochondria reside in (table 2). Considering the large volume of mitochondria needed to meet the demands of cardiomyocytes, fission, fusion, synthesis, and autophagy are very important for efficient energy generation and utilization. To be clear, it is not a fault of the authors of this review to not have all the information; the authors well point out the need for future studies to get more answers.
This reviewer was miffed by the large number of grammatical errors and, in some instances, a poor selection of words, in the manuscript. Therefore, the manuscript was converted to a word document in order to make suggested edits to improve the readability of the report. This file is attached. Conversion results in misalignment of tables and paragraphs, but it is hoped that the authors can glean out the suggested changes as noted in marked changes.

Author Response
See attached letter

Reviewer 3 Report
The manuscript describes mitochondrial fission, and in particular the role of Drp1 in mitochondrial dynamic with the focus on discovery possible therapies for cardiovascular diseases.
Although the topic is interesting and could merit specific discussion, this review suffers several critical points. The aim’s article is not clear; the review is a general explanation of biochemical pathways and proteins involved in mitochondrial fission process, without a critical discussion of data reported. Only few pages in the last chapter are dedicated to explain the role of mitochondrial fission proteins in cardiovascular diseases. The review scheme is confused and conclusions are controversial and is not clear the review aim.
In addition, very recent reviews on the same topic, (“Novel insights into the involvement of mitochondrial fission/fusion in heart failure: From molecular mechanisms to targeted therapies” Cell Stress and Chaperones (2023) https://doi.org/10.1007/s12192-023-01321-4 ; “Current Understanding of the Pivotal Role of Mitochondrial Dynamics in Cardiovascular Diseases and Senescence” Current Understanding of the Pivotal Role of Mitochondrial Dynamics in Cardiovascular Diseases and Senescence” Front Cardiovasc Med 2022 May 18;9:905072; “The role of mitochondrial fission in cardiovascular health and disease” Nat Rev Cardiol. 2022 Nov;19(11):723-736.) reduce the novelty of this review.
In my opinion this review gives no innovative contribution in mitochondria literature. I suggest the authors to reduce the descriptive part of mitochondrial fission process, focusing intead on the new insights into mitochondrial fission proteins interaction networks and their role in cardiovascular diseases.
Author Response
See attached letter

Round 2
Reviewer 1 Report
1. The authors replaced several references as pointed out previously.
2. The figures were made by "Biorender" without proper indication.
Author Response
Ms. Codruta Cormos
Assistant Editor, MDPI Romania
Str Avram Iancu 454, Floresti, Cluj, Romania
Dear Cormos,
Thank you for reviewing the attached manuscript titled “Drp1 mediated mitochondrial fission protein interactome: as an emerging core player in mitochondrial dynamics and cardiovascular disease therapy” [Manuscript ID: ijms-2261523]. We thank the reviewers for all their valuable and critical
comments which helped to direct, focus and improve our manuscript. We have addressed all the reviewers’ comments. We believe this manuscript is improved and submit it for publication in International Journal of Molecular Sciences.
Thank you for your consideration.
Nir Qvit
- The authors replaced several references as pointed out previously.
Response: We thank the reviewer for the review.
2. The figures were made by "Biorender" without proper indication.
Response: We thank the reviewer for the review, we added the sentence "The figure was created with BioRender.com." in the legend of Figure 8.
Reviewer 3 Report
The authors have carefully addressed my concerns and the manuscript has much improved
Author Response
Ms. Codruta Cormos
Assistant Editor, MDPI Romania
Str Avram Iancu 454, Floresti, Cluj, Romania
Dear Cormos,
Thank you for reviewing the attached manuscript titled “Drp1 mediated mitochondrial fission protein interactome: as an emerging core player in mitochondrial dynamics and cardiovascular disease therapy” [Manuscript ID: ijms-2261523]. We thank the reviewers for all their valuable and critical
comments which helped to direct, focus and improve our manuscript. We have addressed all the reviewers’ comments. We believe this manuscript is improved and submit it for publication in International Journal of Molecular Sciences.
Thank you for your consideration.
Nir Qvit
- The authors have carefully addressed my concerns and the manuscript has much improved.
Response: We thank the reviewer for the review.